# Fluorometric Detection of SARS-CoV-2 Single-Nucleotide Variant L452R Using Ligation-Based Isothermal Gene Amplification

**DOI:** 10.3390/bioengineering10101116

**Published:** 2023-09-23

**Authors:** Kangwuk Kyung, Jamin Ku, Eunbin Cho, Junhyung Ryu, Jin Woo, Woong Jung, Dong-Eun Kim

**Affiliations:** 1Department of Bioscience and Biotechnology, Konkuk University, Seoul 05029, Republic of Korea; 2Department of Emergency Medicine, Kyung Hee University College of Medicine, Kyung Hee University Hospital at Gangdong, Seoul 05278, Republic of Korea

**Keywords:** SARS-CoV-2, single-nucleotide mutation, strand displacement amplification, rolling circle amplification

## Abstract

Since the severe acute respiratory syndrome coronavirus 2 (SARS-CoV-2) variant was first discovered, several variants showing different infectivity and immune responses have emerged globally. As the conventional method, whole-genome sequencing following polymerase chain reaction (PCR) is currently used for diagnosis of SARS-CoV-2 mutations. However, these conventional PCR-based direct DNA sequencing methods are time-consuming, complicated, and require expensive DNA sequencing modules. Here, we developed a fluorometric method for the accurate detection of a single missense mutation of U to G in the spike (S) gene that changes leucine to arginine (L452R) in SARS-CoV-2 genomic RNA. Our method for the detection of single-nucleotide mutations (SNM) in the viral RNA genome includes RNA sequence-dependent DNA ligation and tandem isothermal gene amplification methods, such as strand displacement amplification (SDA) and rolling circle amplification (RCA) generating G-quadruplex (GQ). In the presence of SNM in the viral RNA, ligation of both ends of the probe DNAs occurs between 5′-phosphorylated hairpin DNA and linear probe DNA that can discriminate a single base mismatch. The ligated DNAs were then extended to generate long-stem hairpin DNAs that are subjected to the first isothermal gene amplification (SDA). SDA produces multitudes of short ssDNA from the long-stem hairpin DNAs, which then serve as primers by annealing to circular padlock DNA for the second isothermal gene amplification (RCA). RCA produces a long stretch of ssDNA containing GQ structures. Thioflavin T (ThT) is then intercalated into GQ and emits green fluorescence, which allows the fluorometric identification of SARS-CoV-2 variants. This fluorometric analysis sensitively distinguished SNM in the L452R variant of SARS-CoV-2 RNA as low as 10 pM within 2 h. Hence, this fluorometric detection method using ligation-assisted tandem isothermal gene amplification can be applied for the diagnosis of SARS-CoV-2 SNM variants with high accuracy and sensitivity, without the need for cumbersome whole-genome DNA sequencing.

## 1. Introduction

Since the D614G mutation in SARS-CoV-2 was first discovered in Europe in February 2020, many variants that evade host immune responses have emerged [1,2,3]. Some variants of SARS-CoV-2 with missense mutations of the receptor binding domain (RBD) exhibit heightened binding affinity towards the host entry receptor angiotensin-converting enzyme 2 (ACE2), leading to elevated viral infectivity and evasion of host immunity [2]. A missense mutation with a single nucleotide change causes several variants in SARS-CoV-2 lineages, such as L452R, T478K, E484Q, D614G, and P681R in the RBD of spike protein. Of these, L452R (U1355G in the S gene of RBD) observed in mutant variants B.1.427/429 and B.1.617 in 2021, which has been associated with a reduction in the natural/vaccine-induced immune response, has been reported in other globally circulating lineages [3,4,5,6]. Thus, a sensitive and accurate diagnosis of SARS-CoV-2 variants at an early diagnostic stage is needed for thorough surveillance of globally circulating SARS-CoV-2 strains.

Currently, polymerase chain reaction (PCR) with whole-genome sequencing is the main method used for conventional diagnosis of SARS-CoV-2 mutations. However, these conventional RT-PCR and subsequent direct DNA sequencing methods are time-consuming (>2 h or a day for DNA sequencing), complicated, and require expensive apparatus, with significant limitations in resource-constrained environments. Notable challenges include the requirement for advanced thermal cycling machinery and DNA sequencing modules [4,5,6,7,8]. To address these challenging issues of variant diagnosis method, isothermal amplification methods for viral gene amplification have emerged as alternatives to conventional RT-PCR, including loop-mediated isothermal amplification (LAMP) [9], recombinase polymerase amplification (RPA) [10], strand displacement amplification (SDA) [11], and rolling circle amplification (RCA) [12]. Isothermal gene amplification is attractive because of its ability to amplify target genes without temperature variation. Owing to this advantage, isothermal amplification is well suited for situations requiring point-of-care testing (POCT) in resource-constrained environments.

LAMP can detect as few as several copies of DNA in a reaction mixture, with high sensitivity. This method also shows high specificity, efficiency, and rapidity under isothermal conditions [9]. However, it requires a set of four specially designed primers, which is inadequate for the detection of single-nucleotide mutations (SNM). Moreover, the LAMP reaction is performed at a relatively high temperature of 60 °C, which is why it is difficult to integrate it with other moderate isothermal amplification techniques.

RPA is achieved by binding opposing oligonucleotide primers to the template DNA and extending them with DNA polymerase. RPA uses recombinase–primer complexes to scan double-stranded DNA and facilitate strand exchange at specific sites. Unlike conventional amplification methods, such as polymerase chain reaction (PCR), RPA does not require global melting of the template and is carried out isothermally at 37 °C.

SDA has been used for the detection of target genes through target gene amplification using the sequence-specific DNA nicking enzyme (NE) and the exonuclease-deficient Klenow fragment (KF) [11,13]. During the SDA procedure, NE generates a nick by cleaving one strand of the target double-stranded DNA at the recognition site, and KF then extends the DNA from the 3′ end of the nick and displaces the downstream strand. Through perpetual cycling of these enzymatic reactions, target nucleic acids are amplified as multitudes of single-stranded DNAs (ssDNA) that report the presence of target nucleic acids, which can be subsequently utilized as primers or templates in downstream workflows [14,15,16,17,18,19].

RCA is one of the robust isothermal amplification tools, which were extensively employed by many research groups to amplify target nucleic acids, including miRNA, mRNA, and viral RNA [18,19,20,21]. The RCA process amplifies target nucleic acid as multitudes of ssDNA, in which circular padlock DNA annealed with short DNA primer with free 3′-end is served as a template for strand displacing phi29 DNA polymerase to synthesize a long stretch of ssDNA with specific repetitive sequences. Ligation of a linear padlock DNA with splint target nucleic acid sequence is a critical target decision step in most of the RCA-based target gene detection methods. However, this ligation process is likely to be a bottleneck because the alignment of target sequences with padlock DNA is often hindered by factors such as the limited concentration of target molecules, intramolecularly structured nucleic acids, or nonspecific interactions [17]. Thus, this ligation step is often hampered in application to direct detection of long RNAs (>3000 nucleotides; nt), such as viral RNA [22,23]. Thus, use of linear padlock DNA for ligation is very challenging for application to target nucleic acid with long length. To avoid difficulties in ligation of padlock DNA in recognition of long target nucleic acid, our group previously utilized pre-ligated circular padlock DNA for detection of large-sized genes [16,24]. The resulting RCA products can be easily detected with the G-quadruplex (GQ) structure formed in the long stretch of ssDNA with repeating sequences and a fluorescent dye that intercalates the GQ-DNA structure [25].

Previously, we developed a fluorometric detection system utilizing RCA to repetitively generate a G-quadruplex structure (GQ-RCA) and Thioflavin T (ThT) [26,27]. In addition, we have established a fluorometric system for selective and sensitive detection of SNMs present in the leukemogenic fusion gene [28]. Through tandem gene amplification by RT-PCR and subsequent ligation-based GQ-RCA, multiple drug-resistant SNMs in the fusion gene were fluorescently detected. As for SNM detection, this method used ligation between the amplicon DNA and padlock probe DNA to discriminate a single base mismatch. Despite usefulness in detection of SNM in the leukemogenic genes, our previous method requires PCR-based gene amplification and removal of one strand of amplicon DNA to generate ssDNA for annealing to padlock DNA and subsequent ligation. Thus, it is tempting to overcome the limitation of ligation-based RCA by using pre-ligated circular DNA and ssDNA primers. Taking advantage of another isothermal gene amplification method such as SDA, multiple rounds of stand displacement reaction would generate multitudes of ssDNAs that can be utilized in the subsequent RCA. Thus, combining these two isothermal amplification methods, SDA and RCA, was attempted in our study for fluorometric detection of SNM in the target gene through GQ-RCA following the ligation reaction specific for the single-nucleotide variant.

In this study, we developed a fluorometric method for accurate detection of a single missense mutation of L452R in SARS-CoV-2 RNA through tandemly combined isothermal gene amplification tools without the need for direct DNA sequencing. Depending on the presence of mismatched bases between the model viral RNA and linear probe DNA, ligation of both ends of probe DNAs was evaluated. In the presence of L452R mutant RNA, the linear form of probe DNAs was ligated and extended to form a hairpin-type DNA (shown in Figure 1), which was used as substrate DNA for subsequent tandemly combined isothermal gene amplification reactions, SDA, and following GQ-RCA. The amplified ssDNA harboring G-quadruplex was fluorescently visualized and quantified using Thioflavin T fluorophore. This fluorometric detection method combined with ligation-based tandem gene amplification will be useful for sensitive and accurate detection of SARS-CoV-2 variants with SNMs. Using this method, it is possible to detect the SNM of SARS-CoV-2 RNA as low as 10 pM within 2 h, which is comparable to the amount and time required for conventional qPCR. Compared to currently used DNA sequencing for SNM detection following qPCR amplification of viral genes, our method does not need cumbersome direct DNA sequencing by virtue of tandem isothermal gene amplification followed by SNM-specific ligation reaction.

## 2. Material and Methods

### 2.1. Oligonucleotides, Enzymes, and Chemical Reagents

RNA oligonucleotide (53 nts) containing a single missense mutation (U to G) L452R, as well as RNA oligonucleotide of the same sequence devoid the SNM (53 nts) in the wild-type SARS-CoV-2 viral RNA, was chemically synthesized and purified by high-affinity purification (Integrated DNA Technologies, Coralville, IA, USA). DNA oligonucleotides used for the target RNA recognition, such as 5′-phosphorylated hairpin probe (HP, 62 nts) and linear probe (LP, 47 nts), and isothermal gene amplification (RCA and SDA), such as model SDA product (SDAP, 22 nts) and 5′-phosphorylated dumbbell padlock DNA (94 nts), were chemically synthesized and purified by high-performance liquid chromatography (Bionics, Seoul, Korea). Detailed DNA/RNA oligonucleotide sequences are shown in Appendix A. T4 DNA ligase, 10 × T4 DNA ligase buffer, exonuclease I, exonuclease III, and deoxynucleotide mixture (dNTPs) were purchased from Takara Korea Biomedical, Inc. (Seoul, Republic of Korea). For Klenow fragment (exo-), 10 × NEBuffer™2, phi29 DNA polymerase, and nicking endonuclease (Nb.BbvCI) were purchased from New England Biolabs (Ipswich, MA, USA). SYBR Gold dye was purchased from Invitrogen (Carlsbad, CA, USA). Thioflavin T was purchased from Sigma-Aldrich (St. Louis, MO, USA).

### 2.2. SNM-Specific Ligation

To generate a matched base-specific ligated DNA, RNA-splinted DNA ligation between two probe DNAs, hairpin probe and linear probe (HP and LP), was performed in the presence of the target model RNAs (L452R RNA and wild-type RNA). A total of 8 μL of reaction mixture containing 1 × T4 DNA ligase buffer (30 mM Tris-HCl, pH 7.8, 10 mM MgCl_2_, 10 mM DTT, and 10 mM ATP), 0.1 μM of each HP, LP, and model RNA (either wild-type or L452R mutant) was preheated at 95 °C for 3 min. After slowly cooling to room temperature, the reaction mixture was supplemented with 35 U of T4 DNA ligase (350 U/μL) in the T4 DNA ligase buffer. The resulting 10 μL of reaction mixture was incubated at 25 °C for 1 h, and the reaction was quenched by heating at 90 °C for 10 min. The ligation reaction products were analyzed by 10% (*w/v*) urea PAGE and visualized using a UV transilluminator after staining with SYBR Gold.

### 2.3. SDA Reaction

The 3′-end of the resulting ligated DNA or HP probe DNA present in the SNM-specific ligation reaction was extended by Klenow fragment (KF) DNA polymerase used in the SDA reaction, resulting in a long-stem hairpin-type DNA product that was used for the same SDA reaction. The SDA reaction mixture (25 μL) contained 10 μL of the SNM-specific ligation reaction products (wild-type or L452R mutant), 1 × NEBuffer™2 buffer, 5 U of KF, and 10 U of NE (Nb.BbvCI). The reaction mixture was incubated at 37 °C for 30 min and the reaction was quenched by heating at 95 °C for 10 min. During this process, the NE forms the nick of the ligated product, providing a site for KF to be active. As KF is an exonuclease-deficient fragment of DNA polymerase I, it facilitates strand displacement amplification. The SDA products were analyzed by 10% (*w/v*) denaturing PAGE (8 M urea) and visualized using a UV transilluminator after staining with SYBR Gold.

### 2.4. Pre-Ligated Dumbbell Padlock DNAs

To ligate two termini of dumbbell padlock DNA, 3 μM of 5′-phosphorylated linear dumbbell padlock DNA (94 nts) in 1 × T4 DNA ligase buffer was supplemented with 175 U of T4 DNA ligase in a total 10 μL of reaction mixture. After the reaction mixture was incubated at 22 °C for 1 h, the reaction mixture was supplemented with 5 μL of a mixture containing 15 U of exonuclease I, 100 U of exonuclease III, and 1 × exonuclease I buffer (67 mM glycine-KOH, pH 9.5, 6.7 mM MgCl_2_, and 1 mM DTT) and incubated at 37 °C for 2 h to degrade non-ligated linear DNAs. The reaction was terminated by heating the mixture at 95 °C for 10 min. The ligated and exonuclease-treated products were analyzed by 10% (*w/v*) denaturing PAGE (8 M urea) and visualized using a UV transilluminator after staining with SYBR Gold. The resulting reaction aliquot was used as dumbbell padlock DNA in closed form for subsequent RCA reactions.

### 2.5. GQ-RCA and One-Pot SDA/GQ-RCA with Fluorescence Analysis

G-quadruplex-generating RCA (GQ-RCA) reaction was carried out with 100 nM of model SDAP (22 nts) or 5 μL aliquots of the SDA reaction following the SNM-specific ligation reaction with either wild-type or L452R mutant in the GQ-RCA reaction mixture (25 μL). The GQ-RCA mixture contained 200 nM of the pre-ligated dumbbell padlock DNA, 115 U of phi 29 DNA polymerase, 1 × phi29 DNA polymerase buffer (50 mM Tris-HCl, pH 7.5, 10 mM MgCl_2_, 10 mM (NH_4_)_2_SO_4_, and 4 mM DTT), 0.5 mM of dNTPs, 2 mM of KCl, and 15 μM of ThT. The reaction mixture was incubated at 32 °C for 30 min and the reaction was quenched by heating at 95 °C for 10 min.

For the one-pot SDA/GQ-RCA reaction, 25 μL reaction mixtures containing either 100 nM of model SDAP (22 nts) or 9.5 μL aliquots of the SDA reaction following the SNM-specific ligation reaction performed with either wild-type or L452R mutant RNA at various concentrations (1 fM~100 nM), 200 nM of the dumbbell padlock DNA, 5 U of KF, 10 U of NE, 115 U of phi 29 DNA polymerase, 0.5 mM of dNTPs, and 15 μM of ThT in a reaction buffer (50 mM potassium acetate, 20 mM Tris-acetate, 10 mM magnesium acetate, 100 μg/mL BSA, and 2 mM KCl; pH 7.9) were used. The reaction mixture was incubated at 32 °C for 30 min and the reaction was quenched by heating at 95 °C for 10 min.

To visualize the enhanced ThT intercalation into the GQ, the heat-quenched reaction products in a transparent PCR tube, after cooling at 4 °C for 5 min, were photographed under UV illumination using a PowerShot A640 digital camera (Canon Inc., Tokyo, Japan). The fluorescence intensity of each reaction sample (25 μL) containing ThT intercalated into GQ, which was diluted with the same volume of double-distilled water, was measured using a fluorescence spectrophotometer (Cary Eclipse, Agilent Technologies, Santa Clara, CA, USA); λ_ex_ = 425 nm and λ_em_ = 488 nm; PMT voltage = 600. These procedures have been validated through three repetitions, and reproducibility has been confirmed.

## 3. Results and Discussion

To achieve the sensitive and accurate detection of SNM in the SARS-CoV-2 viral genes, we try to design and develop a fluorometric method implementing sequence-specific ligation and tandem gene amplification methods. To this end, a single mutation, L452R, in the SARS-CoV-2 RNA was chosen as a model for SNM detection through tandem isothermal gene amplification tools without direct DNA sequencing.

### 3.1. Strategy of the Ligation-Based Tandem Gene Amplification for Detection of SNM

As depicted in Figure 1, our strategy to detect an SNM in SARS-CoV-2 viral RNA encompasses three consecutive steps: (i) single-base selective DNA ligation between two probe DNAs annealed to the target RNA sequence, (ii) strand extension followed by SDA with NE and KF, and (iii) GQ-RCA permitted with RCA primers generated from the preceding SDA reaction. Accumulation of ssDNAs harboring GQ was fluorescently visualized and quantified using a ThT fluorophore that can selectively intercalate into GQ structures and emit strong fluorescence [21]. Thus, production of the hairpin structure DNA through ligation of two probe DNAs (HP and LP) is critical for subsequent tandem gene amplification, such as SDA and RCA, allowing the detection of SNMs in the target gene. In contrast, in the absence of mutant RNA (i.e., wild-type gene), the tandem gene amplification was not initiated because of the failure of ligation between two probe DNAs, after which an increase in ThT fluorescence was not observed.

As for the ligation reaction, the 5′-end phosphorylated HP was designed to anneal to sequences of both SARS-CoV-2 wild-type and mutant genes, in which the probe DNA binds to the 5′-side of the target sequences (28 bases) adjacent to the single-base mutation site (i.e., L452R sequence). To discern the single-base change (U → G) in the viral RNA, LP was designed to be selectively hybridized to the mutant gene with high discrimination of mismatched bases. The 3′-end position of LP was completely matched and hybridized to the SNM site, resulting in ligation of two ends of the probe DNAs by T4 DNA ligase. Subsequently, the 3′-end termini of the ligated product of the two probe DNAs are extended and the resulting extended hairpin DNA is displaced from the RNA by KF DNA polymerase. The long-stem hairpin DNAs that are generated from ligation of two probe DNAs (HP and LP) and subsequent strand extension encounter NE and KF for SDA, in which multiple oligonucleotide-DNAs (22-mer), as primers for the subsequent RCA, are produced by the repeated action of NE and KF during the SDA reaction. As the RCA primers anneal to the dumbbell padlock DNA, phi29 DNA polymerase initiates the RCA to generate a long stretch of ssDNA containing multiple repeated G-quadruplex structures that can be intercalated by the fluorophore ThT (i.e., GQ-RCA). The G-quadruplex–ThT complex emits strong fluorescence, which allows this system to quantitatively identify the presence of single-nucleotide L452R mutant in viral RNA. In contrast, the presence of the SARS-CoV-2 wild-type gene does not allow two probe DNAs to be ligated due to mismatched bases between the 3′-end base of LP and the wild-type nucleotide at the SNM site (blue-colored nucleotide in Figure 1). Failure to generate the long-stem hairpin DNA by ligation of two probe DNAs leads to no production of amplified genes from the subsequent SDA/GQ-RCA process, resulting in low fluorescence of ThT.

### 3.2. Validation of Ligation-Based Tandem Gene Amplification for Detection of SNM in RNA

To validate the proposed strategy for the fluorometric detection of a single-nucleotide variant in SARS-CoV-2 viral RNA, we evaluated each step of ligation-based tandem gene amplification using SARS-CoV-2 wild-type and L452R SNM model RNAs as target viral RNAs (i.e., wild-type/L452R model RNA, 53-nts). First, we examined ligation of each probe DNA (HP and LP) that anneals to wild-type or mutant model RNA used as splint RNA. As shown in Figure 2A, ligation reaction at increasing reaction time was carried out with wild-type or mutant RNA as target genes, and the ligation reaction product was analyzed by denaturing PAGE. Ligated probe DNAs (120 nts) were observed as distinctive DNA bands only in the presence of mutant RNA (Figure 2A). This result indicates that an SNM-specific ligated product composed of HP and LP can be generated to distinguish between the wild-type and mutant target genes in the T4 DNA ligase reaction.

After confirmation of the SNM-specific ligated product generation, we tested whether the long-stem hairpin DNA generated through ligation of two probe DNAs was eligible for strand displacement amplification (SDA), yielding multiple copies of single-stranded DNA (ssDNA) that can be used as a primer for subsequent RCA. As shown in Figure 2B, the ligated DNA or HP probe DNA present in the SNM-specific ligation reaction performed in a time-dependent manner was subject to SDA reaction with DNA polymerase and nicking enzyme. The products of SDA reaction were analyzed by denaturing PAGE (Figure 2B). DNA bands generated by the SDA reaction were observed only in the presence of mutant model RNA, in which generated ssDNA showed the same size of 22 nts as compared with the control SDA product (SDAP) DNA. Importantly, incremental accumulation of SDAP was observed at increasing ligation reaction time, suggesting that the amount of ligated products limits the rate of SDAP formation in the SDA reaction.

After measuring the SNM-specific ligated product and confirming the subsequent generation of specific SDAP, we evaluated the validity of GQ-RCA for fluorometric detection of the mutant model RNA by isothermal gene amplification followed by fluorescence enhancement. At first, circularized dumbbell padlock DNA was prepared by ligation of each end of linearized dumbbell padlock DNA without splint DNA, after which exonuclease treatment confirmed the formation of the closed form of each padlock DNA (Figure 3A). Unligated linear padlock DNA was removed from the preparation by exonuclease treatment after ligation reaction. We then evaluated the feasibility of the GQ-RCA for isothermal target gene amplification by monitoring ThT fluorescence enhancement with aliquots of the SDA reaction following the SNM-specific ligation for either wild-type or L452R mutant RNA at increasing ligation reaction time (Figure 3B). Afterwards, the RCA reaction was performed at 32 °C for 30 min, in which pre-ligated dumbbell padlock DNA, phi29 DNA polymerase, and G-quadruplex-intercalating fluorophore ThT were included in the GQ-RCA. When the isothermally amplified gene product generated with the GQ-RCA was analyzed by ThT fluorescence, highly enhanced ThT fluorescence was only observed in reactions with the SDA product obtained through the SNM-specific ligation for L452R mutant RNA as well as the positive control obtained when GQ-RCA was conducted with SDAP. These results indicate that SDA reaction products obtained from the SNM-specific ligation for L452R mutant RNA can be used as primers to initiate subsequent GQ-RCA, the fluorogenic isothermal gene amplification step.

### 3.3. Sensitivity of the Ligation-Based Tandem Gene Amplification

To evaluate the sensitivity of the ligation-based tandem gene amplification system for detection of SNM variant RNA, different concentrations of the SARS-CoV-2 wild-type and SNM variant (L452R) RNA (1 fM to 100 nM) were used to detect ThT fluorescence intensity after the one-pot SDA/GQ-RCA assay (Figure 4). The fluorescence intensity gradually increased with increasing amounts of the SNM variant (L452R) RNA. In contrast, the fluorescence intensity was very low for all wild-type RNA samples regardless of RNA concentration. Thus, the signal-to-background (S/B) ratio (i.e., fluorescence intensity of the mutant compared to that of the wild type) gradually increased with increasing concentration of RNA. An S/B ratio of 1.3 or higher (dashed line in Figure 4) was classified as SARS-CoV-2-positive with statistical significance (*p* < 0.05; *n* = 3), resulting in identification of the SNM variant (L452R) of SARS-CoV-2-positive samples. These results indicate that our ligation-based tandem gene amplification assay system can detect the SNM variant of SARS-CoV-2 RNA at concentrations as low as 10 pM within 2 h.

## 4. Conclusions

In this study, we developed a sequence-discernable ligation-based tandem gene amplification system for fluorometric detection of the single-nucleotide variant of SARS-CoV-2 viral gene. Taking advantage of tandemly combined isothermal gene amplification methods, such as SDA and GQ-RCA, our method can detect the single-nucleotide variant L452R mutant RNA at concentrations as low as 10 pM. The fluorometric RNA detection system based on ligation-based tandem isothermal gene amplification in our study could be improved by testing the full-length viral RNA. Our method based on ligation-assisted tandem isothermal gene amplification can use viral RNA without the need for thermal cycling and DNA sequencing with the entire operating time <2 h, with facile visualization of fluorescence signals under UV light. Our detection method developed in this study is potentially useful for fluorometric detection of various single-nucleotide variants that may occur in SARS-CoV-2 genes in addition to the presence of viral RNA in the sample. Full-length viral RNA would be tested for the detection of single-nucleotide variants with our method, in which lengths and concentrations of HP and LP should be adjusted to enhance sensitivity and specificity of ligation. Compared to the viral RNA fragments currently used in our proof-of-concept studies, use of full-length viral RNA would pose challenges in the recognition of single-nucleotide variants using probe DNAs. As a proof-of-concept method for SNM detection, the ligation-assisted tandem isothermal gene amplification method could be applicable to other viruses as convenient and rapid detection of single-nucleotide variants even in resource-limited environments.

## Figures and Tables

**Figure 1 bioengineering-10-01116-f001:**
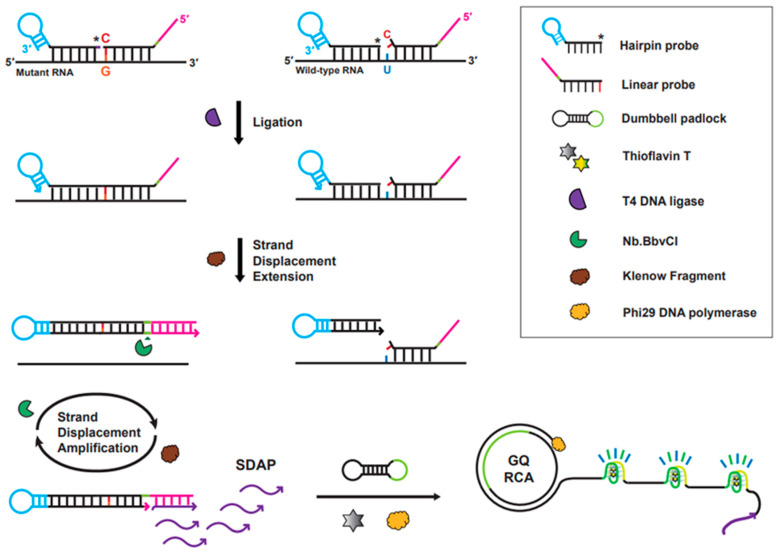
Schematic illustration of fluorometric detection of SARS-CoV-2 variants with a single-base change using ligation-based tandem gene amplification. Asterisk (*) indicates the phosphorylation at the 5′-end of hairpin probe.

**Figure 2 bioengineering-10-01116-f002:**
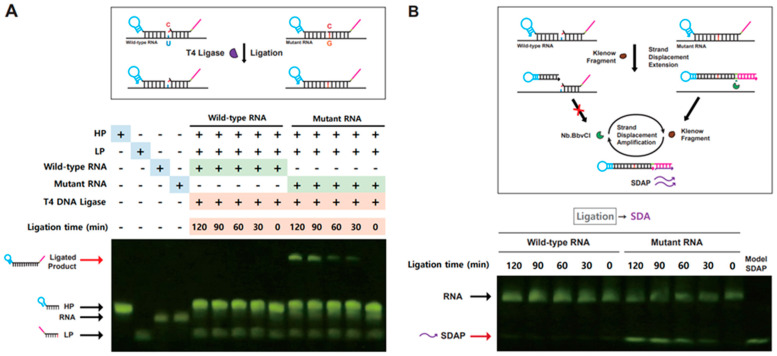
Ligation and strand displacement amplification (SDA) at various ligation times. (**A**) The ligation reaction product formed with HP and LP DNAs at different ligation time points (0 to 120 min), in which the wild-type or mutant RNA was annealed as splint (scheme shown above). The ligation reaction product was analyzed by denaturing 10% PAGE. (**B**) SDA product formed with the ligation reaction product in panel (**A**) (scheme shown above).

**Figure 3 bioengineering-10-01116-f003:**
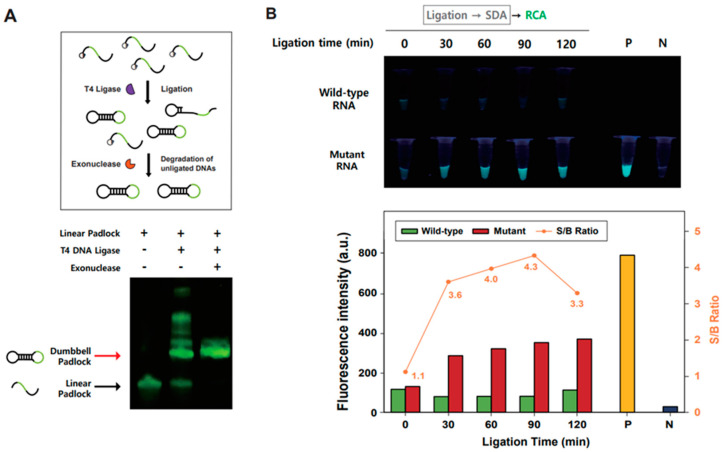
Fluorometric detection of the single-nucleotide mutant RNA with GQ-RCA. (**A**) Electrophoretic analysis of circularization of the dumbbell padlock DNA (scheme shown above). The unligated linear padlock DNA are shown below (black arrow), and the circularized padlock DNAs remain intact after exonuclease treatment (bands shown with red arrow). High-molecular weight DNA bands were disappeared with exonuclease treatment. (**B**) Fluorometric detection of the single-nucleotide mutant RNA with GQ-RCA using ThT fluorescence enhancement. Aliquots of the SDA reaction following SNM-specific ligation for either wild-type or L452R mutant RNA at increasing ligation reaction time were subject to GQ-RCA reaction followed by ThT fluorescence measurement. Each sample used for the ThT fluorescence measurement was visualized under UV light. P indicates the positive control for ThT fluorescence obtained when RCA was conducted with SDAP, while the negative control (N) is the one obtained by RCA reaction performed without SDAP. The bar graph represents ThT fluorescence intensity (λ_em_ = 488 nm) obtained from each RCA product at different ligation times. The orange line represents the signal-to-background ratio (S/B), plotted as the ratio of fluorescence intensity obtained from the mutant to that of the wild type.

**Figure 4 bioengineering-10-01116-f004:**
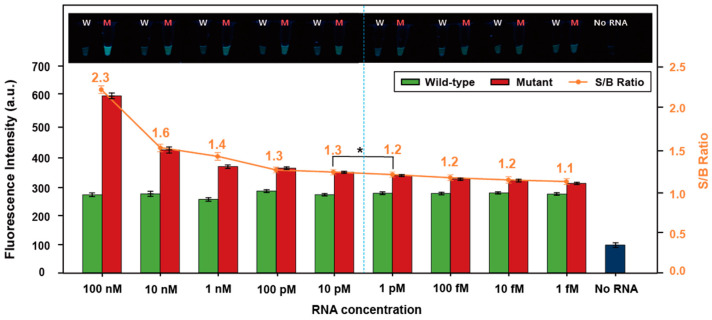
Sensitivity of the ligation-based tandem gene amplification for the detection of the single-nucleotide mutant RNA. Wild-type and mutant RNA at decreasing concentrations (ranging from 100 nM to 1 fM) were tested for fluorometric detection with SDA/GQ-RCA using ThT fluorescence enhancement. The bar graph represents the fluorescence intensity (λ_em_ = 488 nm) obtained after the one-pot SDA/GQ-RCA assay with a different concentrations of the SARS-CoV-2 wild-type and SNM variant (L452R) RNA. Each sample used for ThT fluorescence measurements was visualized under UV light (inset image). The wild-type RNA and mutant RNA are labeled as W and M, respectively. Orange-colored dots represent signal-to-background ratio (S/B) obtained by dividing the fluorescence intensity of mutant RNA by the background fluorescence obtained with wild-type RNA. The samples with 1.3 or higher S/B (left side of dashed line) were statistically classified as SNM variant (L452R) of SARS-CoV-2-positive samples. The experiments were repeated three times (*n* = 3) and error bars display the standard deviation; * *p* < 0.05.

## Data Availability

Data that support the findings of this study are available within the article and Appendix A.

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
