# Peer review of "Fluorometric Detection of SARS-CoV-2 Single-Nucleotide Variant L452R Using Ligation-Based Isothermal Gene Amplification"

_bioengineering, 2023, doi:10.3390/bioengineering10101116_

Round 1
Reviewer 1 Report
This study proposes a fluorometric detection of SARS-CoV-2 single nucleotide mutation using ligation-based isothermal gene amplification. The idea is unique in that it utilizes a sequence of SDA and GQ RCA, the SDA product acts as primers to the GQ RCA, and the detection is done with fluorometric methods. The authors showed a possibility to detect single nucleotide mutation up to 10 pM within 2h. The proposed method could be very useful in resource-constrained environments since it does not require whole-genome sequencing.
There are some comments and questions as follows.
(1) The introduction section is written well. It would be more helpful to highlight the uniqueness and contribution of this study explicitly.
(2) (Line 304) The statistical significance is not clear. How many samples were tested and how the statistical significance was decided?
(3) This study shows detection concentration up to 10 pM. It would be necessary to explain the meaning of this range, especially compared with other methods such as whole-genome sequencing.
(4) The proposed method was applied to chemically synthesized samples with several tens of nts. As the authors say, this method could be improved by testing the full-length viral RNA. More explanations on expected issues about testing with viral RNA, such as sequence length, detection sensitivity, and specificity would be helpful.
(5) This study has been focused on mutation detection, but this proposed method also can be used for detecting viral RNAs. An explanation of this possibility would be helpful.
Sincerely,
The reviewer.
Author Response
Please, refer the attached file for responses.

Reviewer 2 Report
The manuscript presents an innovative approach to detect single-nucleotide variants (SNVs) in SARS-CoV-2 viral RNA using a sequence-discernible ligation-based tandem gene amplification system. The authors have described a comprehensive methodology, and I appreciate their effort in elucidating the scientific content. Here are my observations and recommendations for improvement for each section:.
Comments for authors:
Title:
The title accurately reflects the content of the study and is concise and informative.
Abstract:
1. Even with the restrictions of the abstract length, the abstract could include a sentence placing the study within the broader landscape of existing detection methods. This would help readers understand how the proposed method compares to or improves upon current approaches.
Introduction:
1. Consider providing a brief explanation of what the L452R mutation entails and its potential implications. This could help readers not familiar with the specific mutation understand the importance of detecting it.
2. The section thoroughly introduces the existing challenges of conventional diagnostic methods and highlights the need for more efficient alternatives. To enhance clarity, consider breaking down the discussion of isothermal amplification methods (LAMP, RPA, SDA, RCA) into separate paragraphs. This would make it easier for readers to understand each method's purpose and advantages.
3. A stronger justification for choosing SDA and RCA could be provided. What specific advantages do these methods offer in the context of SARS-CoV-2 mutation detection? This could help readers understand the rationality behind the chosen techniques.
4. The section on RCA and challenges with ligation is well-described. However, it might benefit from a concise summary to emphasize the key hurdles faced in this technique. This would help readers grasp the complexity of the method more easily.
5. The development of the proposed method is explained clearly. To enhance clarity, you could consider using subheadings to separate the different steps in the method: ligation, SDA, GQ-RCA, and fluorescence visualization.
6. When introducing specific steps in the method (e.g., hairpin DNA formation), consider providing brief explanations or definitions for non-expert readers.
Methods:
1. The section could benefit from subheadings to delineate the different experimental steps, such as "Oligonucleotide Synthesis and Purification," "SNM-Specific Ligation," "SDA Reaction," "Pre-ligated Dumbbell Padlock DNAs," and "GQ-RCA and One-pot SDA/GQ-RCA with Fluorescence Analysis." Subheadings would help guide readers through the complex procedures.
2. While many abbreviations are defined, some abbreviations are used without prior explanation. Ensure all abbreviations are introduced and explained upon first usage.
3. Consider providing more context or explanation for some of the oligonucleotides and enzymes used. This is especially important for non-standard components, as it aids readers who may not be familiar with these specifics.
4. Ensure consistent terminology is used throughout the section. For instance, "ligation reaction products" in one instance could be referred to as "ligation reaction aliquot" in another. Such variations might confuse readers.
5. It could be valuable to include a brief statement on how the experimental protocols were validated. This could involve discussing any controls used to ensure the reliability and reproducibility of the results.
Results and Discussion:
1. The section effectively describes the methodology and experimental strategy for detecting single nucleotide mutations (SNMs) in the SARS-CoV-2 virus. However, it might be beneficial to start with a brief recap of the research goal and objectives to provide context for readers before diving into the detailed results.
2. The figures are referenced, but including a brief explanation of what each figure depicts within the text would be helpful. This could provide a clearer understanding of the experimental steps and their outcomes.
3. Utilize transition sentences to smoothly guide the reader from one concept or experimental step to another. This can help maintain the coherence of the narrative and improve overall flow.
4. Extend the discussion of the results to provide insight into the broader implications and significance of the findings. Address how this method compares to existing approaches, its potential limitations, and possible applications beyond the current study.
Conclusions:
1. Briefly address any limitations or constraints of the developed method. This could include discussing potential challenges or scenarios in which the method might not perform optimally. Additionally, suggest potential directions for future research or improvements to overcome these limitations.
2. Conclude by discussing the practical implications of the developed method. How could this method impact the field of viral detection, diagnostics, or genetic research? Consider discussing its potential benefits in terms of early detection, monitoring, and response to emerging variants.
Author Response

(The authors gave the same response as above.)

Round 2
Reviewer 2 Report
Thank you commendable efforts